



# Enhanced Predictability of Antarctic Sea Ice through Sea Ice Thickness Assimilation

Nicholas Williams[1], Yiguo Wang[1], and François Counillon[1]

[1]Nansen Environmental and Remote Sensing Center and Bjerknes Centre for Climate Research, Bergen, Norway

**Correspondence:** Nicholas Williams (nicholas.williams@nersc.no)

**Abstract.**

Understanding the mechanisms of Antarctic sea ice variability, as well as its predictability, remains a central challenge in climate modelling due to the sparseness of observations and the complex processes involved. This study assesses how incorporating sea ice thickness (SIT) observations can improve the reanalysis and prediction skills of Antarctic sea ice over
a period long enough to yield robust conclusions. Two 30-year reanalyses are produced using the Norwegian Climate Prediction Model (NorCPM), with and without LEGOS SIT assimilation, and they are used to initialise year-long hindcasts from 1995–2022 beginning in January, April, July, and October. Assimilation of SIT observations improved estimates of Antarctic sea ice trends, seasonal cycle, and interannual variability - particularly in the West Antarctic and the West Pacific, where sea ice is thick and LEGOS SIT is reliable. The integrated ice edge error (IIEE) was also reduced in the reanalysis during the
austral winter and spring, but a degradation was observed during the austral summer. Hindcasts revealed a long SIT memory, with October initialisation resulting in substantial sea ice extent (SIE) skill gains up to 12 months and January initialisation extending prediction skill by 2–3 months in the pan-Antarctic, with strong improvement in the Weddell Sea and the Amundsen-Bellingshausen Seas. The SIE and SIT prediction skill was also improved in the West Pacific during the austral summer and autumn, a region that previously posed a challenge for prediction skill. We show that SIT observations are important for
improving Antarctic SIE predictions, especially for minimum SIE forecasts in the austral summer and at longer lead times.

## 1 Introduction

Antarctic sea ice is critically intertwined with the global climate system. The formation and melting of sea ice modulate the Southern Ocean, with abrupt reductions in Antarctic sea ice linked to changes in atmospheric circulation across the Southern Hemisphere (Zhu and Song, 2022; Rea et al., 2024) as well as to shifts in global ocean circulation (Ferrari et al., 2014). Brine
rejection during sea ice growth strongly influences the formation of Antarctic Bottom Water (Trevena et al., 2008), while regional changes in ice cover can alter ocean–atmosphere exchanges of carbon dioxide (Søren et al., 2011; Shadwick et al., 2021) and affect the prevalence of persistent organic pollutants (POPs) in the ocean (Corsolini and Ademollo, 2022). Over much of the satellite record, Antarctic sea ice showed a small positive trend (Cavalieri and Parkinson, 2008; Comiso et al., 2011; Parkinson and Cavalieri, 2012), in stark contrast to the ongoing decline of Arctic sea ice (Stroeve et al., 2007, 2014). However,
over recent years, Antarctic sea ice extent (SIE) has declined rapidly (Eayrs et al., 2021; Liu et al., 2023). A coordinated



community initiative led to the formation of the Sea Ice Prediction Network–South (SIPN-South) in 2017, which collects forecasts of Antarctic sea ice extent from 1 December to 28 February (Massonnet et al., 2023). Since its inception, the field of Antarctic sea ice prediction has developed rapidly. There is a need for methods that can constrain Antarctic sea ice variability and enable skilful predictions within a full Earth System model, thereby better understanding its downstream implications and

mechanisms that drive its variability.

On seasonal timescales, the primary mechanism underpinning Antarctic SIE predictability lies in ocean heat content (OHC) anomalies during the growth season (Holland et al., 2013; Guemas et al., 2016; Libera et al., 2022; Marchi et al., 2019). This predictive skill weakens during the melt season but re-emerges in autumn, when OHC anomalies from the previous winter return to the surface as the mixed layer deepens. Marchi et al. (2019) further showed that the effectiveness of this

mechanism depends on mixed-layer depth in the model, with deeper mixed layers yielding higher prediction skill. However, this improvement is often associated with deep convective mixing caused by incorrect representations of the Southern Ocean (Marchi et al., 2019). Many studies show that sea ice initialisation using sea ice concentration has a positive impact on Antarctic SIE prediction skill on seasonal timescales (Guemas et al., 2016; Morioka et al., 2019; Bushuk et al., 2021; Xiu et al., 2025). Stratosphere–troposphere coupling is a key mechanism for predicting SIE during austral spring. Anomalies originating in

the stratosphere propagate downward through the troposphere to the surface, modulating mean sea level pressure in western Antarctica. Improved representation of mean sea level pressure in this region subsequently benefits both the dynamical and thermodynamical sea ice evolution during austral spring (England et al., 2016; Wang et al., 2021; Xiu et al., 2025).

The role of SIT in predictability in Antarctica has received little attention due to the paucity of SIT observations. Morioka et al. (2021) investigated the added skill from initialising SIT using a reanalysis dataset from 1986-2017, finding improved

summertime sea ice prediction in the Weddell Sea from wintertime SIT initialisation.Bushuk et al. (2021) initialises the SIT by nudging in an ensemble system temperature, wind fields and humidity towards atmospheric reanalysis data and SST and SIC towards daily OISST data. This ensemble is used to initialise predictions, thereby improving sea ice prediction in the Weddell Sea. With the emergence of SIT observations, there is a perception that SIC and SIT initialisation for Antarctic sea ice prediction (Bushuk et al., 2021; Luo et al., 2023) can enhance prediction skill and improve our understanding of their

individual roles in different regions of Antarctica. Antarctic SIT has been previously assimilated into only a few studies and has been assessed only for its impact on reanalysis. For instance, assimilation of SIC and winter SMOS SIT was implemented by Luo et al. (2021), and found that SIT assimilation improves reanalysis sea ice estimates. They emphasised that careful observation error estimates for Antarctic SIT are needed. Chenal et al. (2024) assimilated radar freeboards from Bocquet et al. (2023) in the winter seasons for both the Arctic and the Antarctic, finding that assimilation improves reanalysis of the Antarctic

sea ice state. The positive impacts are lower in the Antarctic than in the Arctic, but also harder to quantify due to more limited data availability and larger uncertainties in these data. It is also important to assess whether improvements in the reanalysis are reflected in better predictions through a hindcast dataset that is long enough to yield statistically significant results.

There has been substantial work on developing SIT observations over the past 2 decades (Wingham et al., 2006; Laxon et al., 2013; Ricker et al., 2017; Petty et al., 2020; Landy et al., 2022), despite the challenges relating to uncertainty in snow

depth and melt ponds (Sallila et al., 2019). The retrieval of Antarctic SIT, however, presents an even greater challenge than in



the Arctic, since snow depth is thicker and very few historical in-situ records are available (Kaleschke et al., 2024; Bocquet et al., 2024). While Warren et al. (1999) was able to produce a climatological snow depth record for the Arctic using previous in-situ records, this was not possible for the Antarctic. However, in the last few years, two separate products for Antarctic SIT have been produced. Kaleschke et al. (2024) used the Satellite Moisture and Salinity (SMOS) satellite to produce a pan-
Antarctic estimate of thin sea ice. This data is retrieved from the SMOS L-band, and uncertainty in SIT increases with SIT; SIT measurements above 1 metre are too uncertain due to sensitivities in the SMOS L-band. This product is only available during the Austral winter season (April-October) from 2010-present and has been validated primarily in the Weddell Sea (Kaleschke et al., 2024). Additionally, SMOS tends to underestimate Antarctic SIT because it assumes 100% sea ice coverage (Kaleschke et al., 2024). A second Antarctic SIT product has been produced by LEGOS (Bocquet et al., 2024), available year-round in the
Antarctic. This product uses ERS-1, ERS-2, ENVISAT and CryoSat-2 along-track measurements from 1994 to 2023 to produce a sea ice freeboard, which is then converted to SIT using hydrostatic equilibrium. The LEGOS dataset has a higher uncertainty for thinner ice measurements, particularly for ice thinner than one metre. Additionally, a daily Antarctic Ice, Cloud and Land Elevation Satellite (ICESat) and ICESat-2 SIT product has also recently been produced using a deep learning approach (Ma et al., 2025). This data is based on the sparse along-track laser altimetry measurements made by these satellites and is trained on
SIT from ocean reanalyses. This data is minimal during the ICESat period - covering only a few months in each year between 2003 and 2009 due to ICESat operational issues. However, for ICESat-2 there is a consistent period of coverage from October 2018 to the present year-round. The ICESat SIT dataset showed good performance when validated against ULS measurements in the Weddell Sea and compared against other Antarctic SIT observations available (Ma et al., 2025).

This study aims to examine how assimilating SIT observations in the Antarctic over a 30-year period enhances the skill
of Antarctic sea ice predictions. We only assimilate LEGOS SIT, as the SMOS SIT and ICESAT-2 coverage are too short to demonstrate added value for seasonal prediction. However, they are both used for validation. A reanalysis spanning 1994–2023 is produced and used to initialise seasonal hindcasts (4 start dates per year). To our knowledge, this represents the first use of such a long Antarctic year-round SIT record for initialising sea ice predictions in a global climate model. Previous studies examining increased skill from SIT initialisation have only used reanalysis datasets for SIT initialisation. The resulting
reanalysis also provides a more skilled representation of Antarctic sea ice climate variability, strengthening the basis for prediction. We assess both pan-Antarctic and regional impacts on seasonal prediction skill, with the extended observational period enabling a more robust evaluation of the results and their significance.

The paper is organised as follows. In Section 2 we outline the global climate model and observation datasets used in this study. In Section 3 we present the experimental design and metrics used to evaluate the results. In Section 4, we briefly
compare the performance of two 30-year reanalyses, one assimilating ocean and SIC and one assimilating in addition SIT. We then present how SIT can improve sea ice prediction at different target months, lead months and regions by comparing two sets of seasonal hindcasts initialised from the reanalyses. We conclude with a discussion of key results and a summary in Section 5.



## 2 Norwegian Climate Prediction Model

NorCPM is a physics-based global climate model, combining the Norwegian Earth System Model (NorESM, Bentsen et al., 2012) and a deterministic Ensemble Kalman Filter (EnKF, Evensen, 1994; Sakov and Oke, 2008) ensemble data assimilation method. NorCPM is used to produce climate reanalysis (Counillon et al., 2016; Bethke et al., 2021; Wang et al., 2025) and initialise predictions on different timescales (Counillon et al., 2014; Wang et al., 2019; Xiu et al., 2025; Williams et al., 2022). In this Section, we will present the version of NorCPM used in this study, along with the observational datasets used in the data assimilation.

### 2.1 NorESM

In this study, we use the medium-resolution NorESM version 1 NorESM1-ME (NorESM1-ME, Bentsen et al., 2012). NorESM is based on the Community Earth System Model (CESM, Hurrell et al., 2013), but with the ocean and atmosphere components changed. The original Community Atmosphere Model version 4 (CAM4, Neale et al., 2010), is replaced with the atmospheric model CAM4-OSLO with a prognostic aerosol life cycle formulation using emissions and new aerosol-cloud interaction schemes (Kirkevåg et al., 2013). The ocean component is replaced by the Bergen Layered Ocean Model (BLOM), a general circulation model which uses an isopycnal layered coordinate system. The sea ice component is the Los Alamos Sea Ice Model version 4 (CICE, Hunke et al., 2015). The land component is the Community Land Model (CLM) version 4 (Lawrence et al., 2011). The model components are coupled using version 7 of the CESM coupler (Craig et al., 2012).

The atmosphere and land models have a horizontal resolution of 1.9° in latitude and 2.5° in longitude and 26 hybrid sigma-pressure vertical levels. The ocean and sea ice models have a horizontal resolution of $1° \times 1°$. BLOM includes 51 isopycnal layers and a bulk mixed layer, with time-evolving thicknesses and densities. NorESM runs with CMIP5 historical forcings (Taylor et al., 2012) and the Representative Concentration Pathway 8.5 (RCP8.5) (Moss et al., 2010) beyond 2005.

The ice thickness distribution in CICE is represented using five thickness categories, which is optimal for representing the sea ice cover at reasonable computational cost and speed (Bitz et al., 2001; Massonnet et al., 2019). The thermodynamic equations are solved by the one-dimensional vertical Bitz and Lipscomb model (Bitz and Lipscomb, 1999), including parametrisations for melt pond (Hunke et al., 2013), aerosol (Holland et al., 2012) and radiation transfer parameterisations (Briegleb and Light, 2007). The incremental remapping scheme of Lipscomb and Hunke (2004) is used to solve for the horizontal transport of sea ice. The sea ice stresses are formulated using the elastic-viscous-plastic rheology (Hunke and Dukowicz, 1997). The remapping scheme of Lipscomb (2001) transports the sea ice in thickness space.

### 2.2 Assimilation implementation

NorCPM uses a sequential data assimilation method characterised by a Monte Carlo update followed by a linear analysis update (Evensen, 1994). Specifically, we use a deterministic formulation of the EnKF (Sakov and Oke, 2008), which solves the analysis without needing to perturb the observations. This method is multivariate, and all or any of the model state vector (depending on user choice) can be updated through their ensemble covariances with the observations. The deterministic EnKF




is particularly useful for many applications as it is more computationally efficient and outperforms the standard EnKF for smaller ensemble sizes.

In this study, we use monthly assimilation, assimilating monthly averages of observations on the $15^{th}$ day of each month. The model state is instantaneously updated based on all observations (Counillon et al., 2014). This update is split into two steps: firstly, ocean model variables and the multi-category ice concentration per unit grid cell are updated based on the anomaly assimilation of the ocean and SIC variables. In the second step, the multi-category ice concentration per unit grid cell is updated with full-field assimilation of SIT observations.

In NorCPM, anomaly assimilation is employed for ocean and sea ice concentrations because the ocean and sea ice concentrations (that are tightly linked) are strongly attracted to their bias state, leading to forecast drift and a transfer of bias into non-observed quantities via ensemble covariance (Carrassi et al., 2014; Counillon et al., 2016). Unlike for sea ice concentration, the SIT bias can take up to a decade to re-emerge (Bethke et al., 2021). Therefore, we use full-field assimilation in this study to correct the SIT bias, which, in turn, can influence the system's variability. This two-step method of anomaly and full-field assimilation has already been employed successfully and produces stable performance (Williams et al., 2025).

The assimilation of ocean temperature and salinity profiles, sea surface temperature (SST), SIC and SIT observations is performed as described in Williams et al. (2025), though the observational datasets for SST, SIC and SIT differ. Strongly coupled ocean-sea ice data assimilation is performed, meaning that both ocean and sea ice observations are used to update both components (Kimmritz et al., 2018). The full ocean model state vector is updated in isopycnal coordinates (Counillon et al., 2016) and the full multi-category ice concentration is updated. However, the sea ice volume (SIV) per unit grid cell ("vicen" in CICE) is estimated so that the thickness of each thickness category remains identical to that of the prior. This prevents the need to reshuffle ice to a different thickness category in the post-analysis, which counteracts the emergence of a drift and has been shown to be optimal in an idealised twin experiment (Kimmritz et al., 2018). Similarly, the energy budget for each multi-category sea ice quantity is recomputed (Kimmritz et al., 2018).

To handle sampling errors that can cause significant problems in ensemble-based data assimilation systems – like the one we use in NorCPM– we use a series of ad-hoc techniques that are available and widely tested in NorCPM (Counillon et al., 2016; Bethke et al., 2021). Firstly, an R-factor is used (Sakov et al., 2012). It inflates the observation error by a factor (of 3 in NorCPM, see Bethke et al. (2021) ) for the update of the ensemble anomaly, which prevents the spurious deflation of the ensemble spread. The K-factor scheme (Sakov et al., 2012) inflates the observation error so that the analysis remains within a certain multiple of the ensemble standard deviation (we use 2 in this study) and prevents the risk associated with overly strong updates (e.g., from erroneous observations, or a model not able to cope with real variability). Lastly, we use localisation (Evensen, 1994), whereby only local observations are used in the analysis update to limit spurious long-range correlations. We define a localisation radius of 800 km for the sea ice model (Kimmritz et al., 2019; Bethke et al., 2021; Williams et al., 2025) and use a localisation radius dependent on latitude for the ocean (Wang et al., 2017) –varying between 1500 and 2300 km. A smooth distance-weighted Gaspari and Cohn function (Gaspari and Cohn, 1999) is used to ensure continuity in the analysis update.





In this study, we assume that the observation errors are independent of each other. The observation errors of the SST and SIC
products may, however, be correlated due to the post-processing in the production of the final gridded product. We mitigate
this in NorCPM by only using the nearest observation of SST, SIC and SIT respectively in the local analysis (Counillon et al.,
2016; Kimmritz et al., 2019). All hydrographic profile data within each local window are used (Wang et al., 2017).

## 2.3 Datasets for assimilation

Hydrographic data profiles of temperature and salinity are taken from the EN4.2.2 dataset (Good et al., 2013). The data in
this dataset are split into different categories depending on their quality (Gouretski and Cheng, 2020). We use only the highest
quality data (category 1) in this study. The observation errors used are as in Wang et al. (2017), accounting for instrumental
and representation error.

SST and SIC monthly average data are from the UK Met Office operational sea surface temperature and sea ice analysis
(OSTIA) dataset (Donlon et al., 2012). All the satellite data used in this dataset is sourced from the EUMETSAT's Ocean
and Sea Ice Satellite Application Facility (OSISAF) (Worsfold et al., 2024). SST is sourced from the ESA SST CCI dataset
(Embury et al., 2024), while SIC is sourced from the OSI-450, OSI-430 and OSI-430-b datasets (Lavergne et al., 2019). In-situ
SST data from drifting and moored buoys sourced from the HadIOD version 1.2.0.0 dataset are additionally used to produce
the OSTIA dataset. The SST and SIC data are synchronised for the product, and no SST data is assimilated in grid cells where
sea ice is present. The OSTIA dataset is available on a $0.05° \times 0.05°$ horizontal grid, a much higher resolution than our model
and likely higher than its effective resolution. We have used superobing (Sakov et al., 2012), so that the observation estimate
is the average of all observations within that grid cell, and the error is the harmonic mean of the daily error estimate from the
provider. However, it has been found necessary to inflate the observation error standard deviation by a factor of three, to sustain
good reliability in our ensemble and have an observation estimate within the same ballpark as other products (e.g. HadISST2
used in NorCPM previously, Bethke et al. (2021). Furthermore, we impose that the observation error standard deviation be no
lower than 0.1 °C. SIC observation errors are not provided with the dataset, so we use a 20% constant value (Kimmritz et al.,
2019), which is the generally agreed upon value for the highest uncertainties in the summer melt season (Ivanova et al., 2015;
Cavalieri and Parkinson, 2012; Comiso, 2017).

SIT data are from the LEGOS dataset (Bocquet et al., 2023, 2024). We assimilate both the Arctic and the Antarctic data,
though we do not present any analysis of Arctic results in this study. In the Arctic, where melt ponds obfuscate radar returns,
and the dataset is only available in the winter months (October - April). This problem is reduced in Antarctica, because
snow over sea ice is much thicker there (Istomina et al., 2022), which prevents melt pond formation and its impact on SIT
retrieval. As such, the data is available year-round from April 1994 to June 2023. LEGOS SIT is the first observation dataset
to provide a consistent spatiotemporal resolution and representation of Antarctic SIT covering such a long period. As such, it
represents an ideal dataset for assessing the impact of SIT on reanalysis and prediction over a sufficiently long period to provide
robust conclusions. The LEGOS SIT dataset is derived from along-track radar freeboard from ERS-1, ERS-2, ENVISAT and
CryoSat-2. ERS-1, ERS-2 and ENVISAT used a different, less well-understood mode to retrieve sea ice freeboard compared
with CryoSat-2, which uses synthetic aperture radar. A calibration to correct for biases introduced by the different retrieval





methods is done using a neural network (Bocquet et al., 2023). The observation uncertainty is estimated in Bocquet et al. (2024) using an ensemble method. The main sources of uncertainty in deriving SIT are identified (snow depth, snow density,
ice density, etc.) and multiple sources of in-situ, satellite or modelling estimates of these are used to estimate each of these and produce an ensemble. The ensemble standard deviation is used to quantify the observation uncertainty for the data assimilation.

## 3   Experiments and validation

### 3.1   Experiments

For this study, we have used the NorCPM setup described in Section 2 to produce two new reanalyses and hindcast datasets.
One assimilates ocean and SIC, while the other additionally assimilates SIT. The two reanalyses cover the period from January 1994 to December 2023 and have 30 ensemble members. Both reanalyses are branched on 15 January 1994 from the pre-existing NorCPM reanalysis of Kimmritz et al. (2019), which assimilates temperature and salinity profiles from the EN4.2.2 dataset, SST and SIC from the Hadley Center Sea Ice and SST dataset (Titchner and Rayner, 2014), and is initialised in 1982 from the NorESM ensemble of free runs. A NorESM ensemble of free runs is used in addition, when informative, to identify
part of the variability related to forcings (e.g., anthropogenic, solar, and volcanic). Two seasonal hindcast datasets are initialised from the reanalyses every year from 1995 to 2022, in January, April, July and October of each year, each run for a year. Each hindcast uses 10 ensemble members, chosen from the first 10 ensemble members of the respective reanalyses (all members are equally likely with the EnKF). The only difference between the two reanalyses - and as a consequence the two hindcast datasets - is in the SIT observations assimilated in **EXP-OCT**:

**NorESM**: A free run of the NorESM model. 30 ensemble members are randomly selected from a stable pre-industrial simulation in 1850 and are then integrated until 2023.

**EXP-OC**: Assimilates EN4.2.2 temperature and salinity profiles, SST and SIC data from the OSTIA dataset (Section 2.3) with an anomaly-field assimilation framework.

**EXP-OCT**: In addition to the observations assimilated in EXP-OC, EXP-OCT also assimilates SIT data with full-field
assimilation (Section 2.3).

### 3.2   Validation

We subdivide the Antarctic into five regions (as in Xiu et al. (2025), Figure 1), based on longitude. These are Weddell Sea (60° W - 30° W), Indian (-30° W - 70° E), West Pacific (70° E - 150° E), Ross Sea (150° E - 150° W) and the Amundsen-Bellingshausen Seas (150° W - 70° W). These five regions are chosen due to their distinct geography (particularly differences
in the sea ice cover) and model performance (Xiu et al., 2025).

The reanalysis and hindcasts of SIC and SIE are evaluated against the monthly averaged OSTIA dataset (Section 4). The reanalysis and hindcast SIT are validated against LEGOS SIT, SMOS Antarctic SIT dataset v3.3 (Kaleschke et al., 2024) and





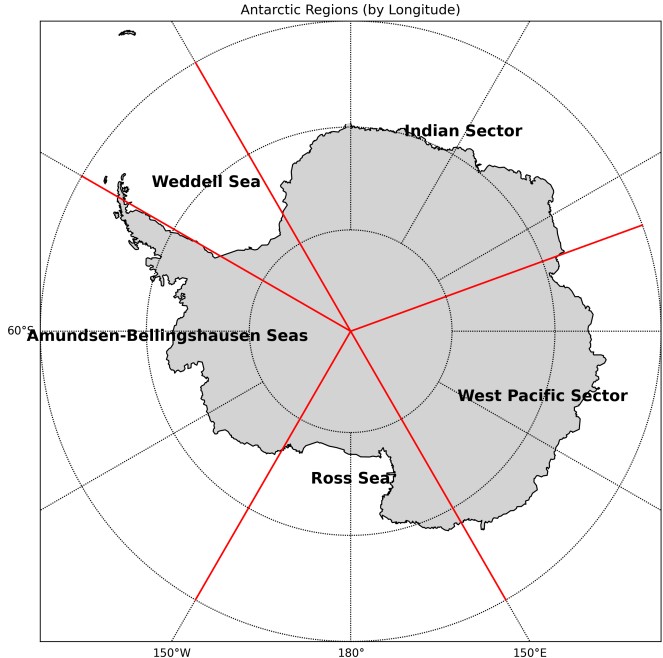

**Figure 1.** Map showing the 5 regions in the Antarctic used in this study. The regions are defined as: Weddell Sea (60° W - 30° W), Indian (-30° W - 70° E), West Pacific (70° E - 150° E), Ross Sea (150° E - 150° W) and the Amundsen-Bellingshausen Seas (150° W - 70° W).

the ICESat deep-learning derived SIT (Ma et al., 2025). Only SMOS SIT below 1 m is used for validation. The ICESat data from 2003–2009 are excluded because missing days in each coverage period result in incomplete sampling for monthly mean 225 calculations.

Various metrics are used to analyse the reanalyses and hindcasts in this study. We define the bias and RMSE as follows:

$$\text{bias} = \sum_{i=1}^{N} W_i(x_i - y_i), \tag{1}$$

$$\text{RMSE} = \sqrt{\sum_{i=1}^{N} W_i(x_i - y_i)^2}, \tag{2}$$

where $N$ is the total number of data points, $W_i$ is a normalising area-based weight constant –i.e. when the metric is computed 230 considering grid cells that do not have the same area. For point-wise bias and bfRMSE calculations, $W_i$ is $\frac{1}{N}$. $x_i$ is the model ensemble mean values, and $y_i$ is the observed values, which are both anomalies from their respective climatology (model or observation). These are usually averaged spatially or temporally (where each grid cell is then weighted by its area).

We also use the mean squared skill score (MSSS), which quantifies the reduction in error relative to the free run. MSSS is defined as

$$235 \quad \text{MSSS} = 1 - \frac{MSE_{forecast}}{MSE_{reference}}. \tag{3}$$





Another benefit of MSSS is that it normalises errors by their pointwise variability and, as such, can help visualise improvements that would otherwise be masked when the amplitude of the error is smaller than in other regions.

The anomaly correlation coefficient (ACC) is also used to test the skill in reproducing the observed variability in the reanalyses and hindcasts. The ACC is defined as

$$\text{ACC} = \frac{\sum_{i=1}^{N} x_i' y_i'}{\sqrt{\sum_{i=1}^{N} x_i'^2} \sqrt{\sum_{i=1}^{N} y_i'^2}}, \tag{4}$$

where $x_i'$ and $y_i'$ are model and observation values. In this study, we always analyse detrended ACC values, i.e., detrending the time series of the experiments and the observations before calculating the ACC. The trend is usually removed because it is considered to be easy to predict (Bushuk et al., 2017). We test the statistical significance of the ACC using a Student's t-test at a 5% significance level, with degrees of freedom calculated as in Von Storch and Zwiers (2002).

The final metric used is the integrated ice edge error (IIEE) (Goessling et al., 2016). This metric is a stronger indicator of sea ice edge location skill than SIE. IIEE is defined as follows:

$$\text{IIEE} = \int_A \max(c_x - c_y, 0) dA + \int_A \max(c_y - c_x, 0) dA, \tag{5}$$

where $A$ is the area, $c = 1$ where the SIC is above 15% and 0 elsewhere, with subscripts $x$ and $y$ denoting the model and observations respectively. IIEE then, is a sum of all areas where SIE is overestimated (first term), and underestimated (second term). IIEE is useful for ice-edge location skill because SIE only accounts for the overall values.

## 4 Results

### 4.1 Reanalysis

We first present results from the two reanalyses, starting with an analysis of the seasonal cycle of SIE and SIV, followed by a detailed analysis of SIT with the assimilated and independent datasets, and followed by an analysis of the SIC trend and then the interannual variability.

The differences in the seasonal cycle of SIE between the two reanalyses are negligible. As we use anomaly assimilation for ocean and SIC, they both show a large bias compared to observations. On the contrary, the seasonal cycle of SIV is strongly different between EXP-OC and EXP-OCT (Figure 2b). Antarctic SIV is increased in EXP-OCT in austral winter, and decreased during the rest of the year in comparison with EXP-OC and NorESM. The timing of the peak volume season also differs in EXP-OCT, as ice volume almost reaches its peak already in August.

The sea ice thickness (SIT) bias is substantially reduced in EXP-OCT (Figure 3) when compared to the assimilated LEGOS product as expected. In both February and September, EXP-OCT exhibits smaller overall biases than EXP-OC, particularly in regions of thick ice in the Weddell and Ross Seas, where the observational data is most reliable. In February, EXP-OCT better reproduces the thin, patchy late-summer ice cover, whereas EXP-OC tends to show thicker SIT in the Weddell and Ross Seas than in observations. In September, when sea ice reaches its maximum extent, both simulations underestimate SIT in the




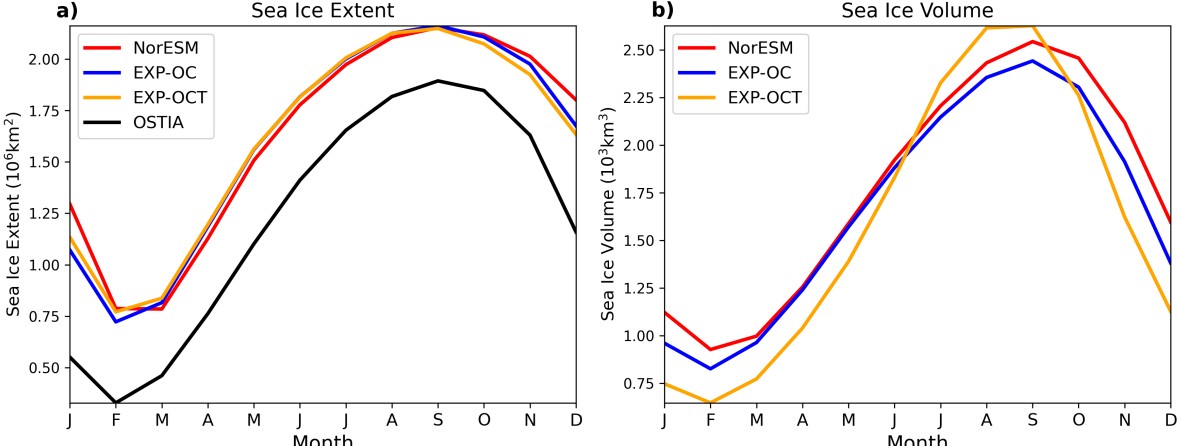

**Figure 2.** Monthly climatologies of the Antarctic sea ice extent (a) and Antarctic SIV (b) in NorESM, EXP-OC, EXP-OCT and the OSTIA observations (only in panel a) over the 1994-2023 reanalysis period.

interior pack but display positive biases along the ice edge. These positive edge biases coincide with areas where the model simulates a higher SIE than observed (Figure 2), suggesting that part of the SIT bias arises from extent mismatches rather than from intrinsic thickness errors. The extent mismatches are a result of positive SST biases present in NorCPM (Bethke et al., 2021; Wang et al., 2025).

We also evaluate the experiments against the SMOS Antarctic thin SIT dataset and the deep-learning derived ICESat SIT dataset, which are independent SIT products. Both datasets are converted into monthly means and re-gridded onto the NorCPM grid for comparison. SMOS retrievals are limited to austral winter months (April–October) and are unreliable for SIT exceeding 1 m due to high uncertainty. Comparisons with ICESat-2, which provides accurate estimates of thicker ice, but covers a more restricted time period, show an interesting complementary analysis.

EXP-OCT exhibits consistently lower RMSEs than EXP-OC and LEGOS when compared with both SMOS and ICESat (Figure 4). This demonstrates that assimilating LEGOS SIT improves NorCPM's representation of SIT, despite substantial differences between LEGOS and SMOS SIT arising from contrasting sensor characteristics and retrieval methodologies (Kaleschke et al., 2024; Bocquet et al., 2024; Chenal et al., 2024). EXP-OCT achieves the lowest RMSEs in austral spring and austral autumn, while errors are higher in February at the end of the melt season, and during the winter, when the ice is thickest and

more variable. EXP-OCT achieves the largest RMSE reduction outside winter when compared to ICESat-2. This likely reflects two factors. Firstly, there are greater uncertainties in the modelled SIT in the transitions between the melt and growth seasons. Second, greater SIT variability and uncertainties in snow depth and density during winter increase noise in the conversion of freeboard to thickness, reducing the assimilation increments where uncertainty is higher.

        SMOS uncertainty increases with ice thickness - becoming substantial above 0.5 m and too large beyond 1 m (Kaleschke

et al., 2024). Therefore, we further examined model skill by SIT ranges (Figure 5). EXP-OCT shows improved performance

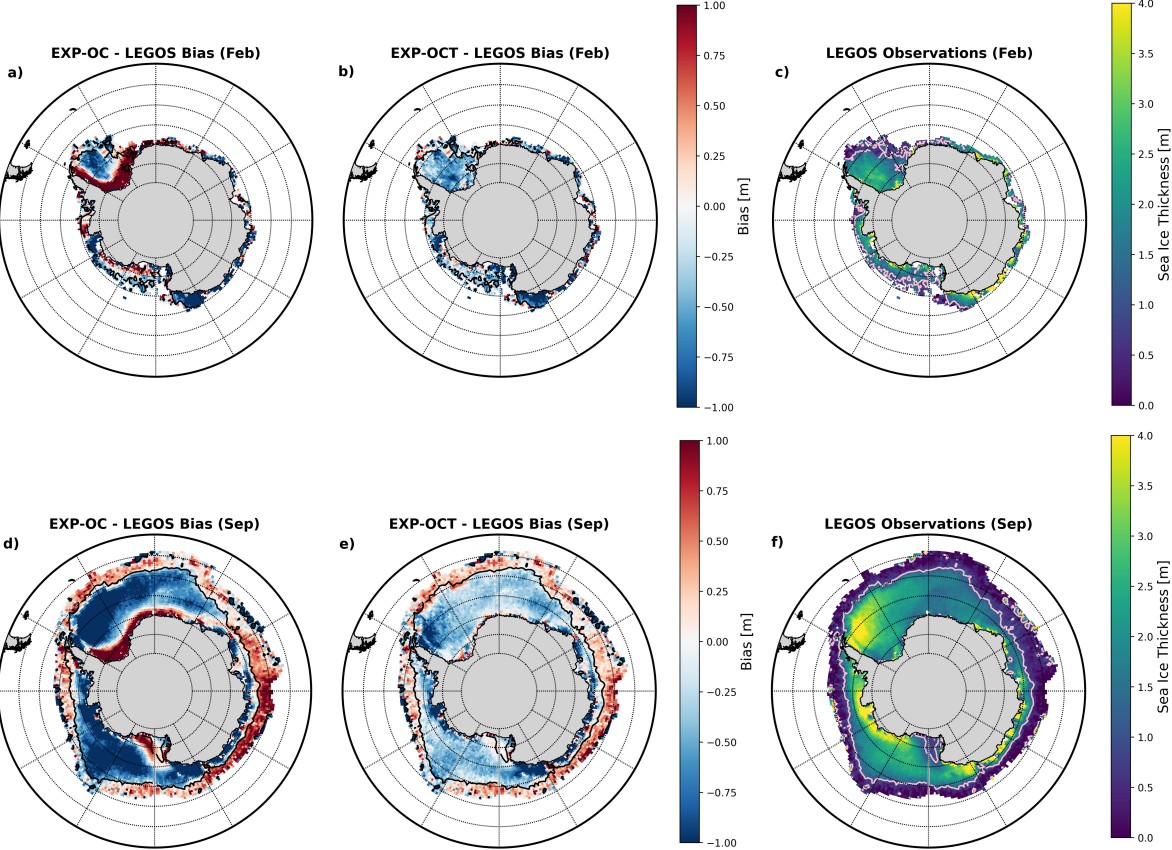

**Figure 3.** Spatial biases in monthly SIT for February and September in EXP-OC and EXP-OCT compared to LEGOS (LHS, middle), and monthly SIT climatology in LEGOS (RHS). The pink contour (reported as black in panel a–d) delimits the 1 m SIT threshold in the LEGOS dataset under which the dataset can be considered less reliable.

across all thickness bins, with the largest reductions in RMSE for ice between 0.2 and 0.8 m. Improvements are therefore not limited to the ranges where SMOS uncertainty is greatest, indicating that the assimilation enhances SIT representation even within the more reliable SMOS measurements below 0.5 m. This suggests that LEGOS SIT assimilation and its observation errors are well formulated, demonstrating again that we do not simply recreate the LEGOS SIT dataset in NorCPM from the

assimilation.

Spatial bias analyses further illustrate how SIT assimilation impacts regional patterns. Comparisons with SMOS (Figure 6), are particularly relevant for thin ice. Both reanalyses overestimate SIT along the sea ice edge and near the continental shelf. EXP-OCT substantially reduces this overestimation in May – when thin ice dominates – in the Indian Sector and the Ross Seas, but yields a slight degradation in the Weddell Sea and parts of the West Pacific Sector. In September, at a time when

SIT is thicker and despite the overall bias being reduced, the bias reduction is only evident in the West Pacific Sector, and a rather strong degradation is found in the Weddell and Amundsen-Bellingshausen Seas. As with SMOS, EXP-OCT reduces





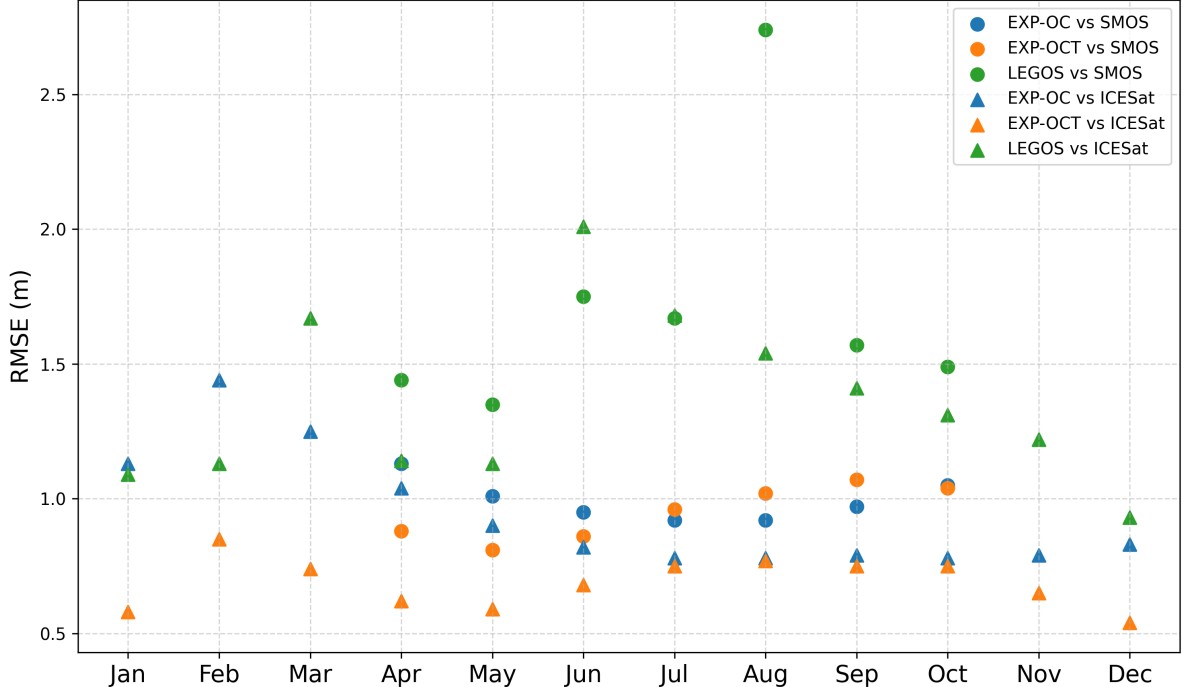

**Figure 4.** Monthly mean sea ice thickness RMSE for EXP-OC, EXP-OCT and LEGOS compared against SMOS and ICESat observations. Circles represent RMSE values versus SMOS between 2010 and 2023, while triangles represent RMSE values versus ICESat between 2019 and 2023.

widespread positive biases most in February relative to EXP-OC, with the largest improvements occurring in the Weddell and Ross Seas near the continental shelf. Improvement remains near the continental shelf in September. As with the SMOS comparison, the bias increases near the ice edge in the Weddell and Ross Seas. This corresponds to the region where we

observed a clear improvement over LEGOS (Figure 3), suggesting that SIT in LEGOS dataset may be too thick in that area.

Overall, the assimilation of LEGOS SIT yields a more balanced and spatially consistent ice thickness distribution, improving both thin ice and thick ice regimes. The cross-dataset agreement with SMOS and ICESat-2 confirms that the assimilation benefits propagate dynamically, producing a more realistic and physically coherent representation of Antarctic SIT. The assimilation of SIT substantially reduces NorCPM's SIT biases and is expected to enhance SIE prediction skill through the

known influence of SIT memory on seasonal predictability, particularly in the Arctic and the Weddell Sea (Bushuk et al., 2020, 2021; Morioka et al., 2021). The consistent RMSE reductions across both independent datasets indicate that assimilating LEGOS SIT improves NorCPM's representation of sea ice conditions rather than merely enforcing the LEGOS SIT variability in our model.

We analyse the capability of the SIT assimilation to improve the SIC trends (Figure 8). EXP-OC does show some agreement

with the observed trend, but misses some important details, such as in the Ross Sea where the observation shows a strong



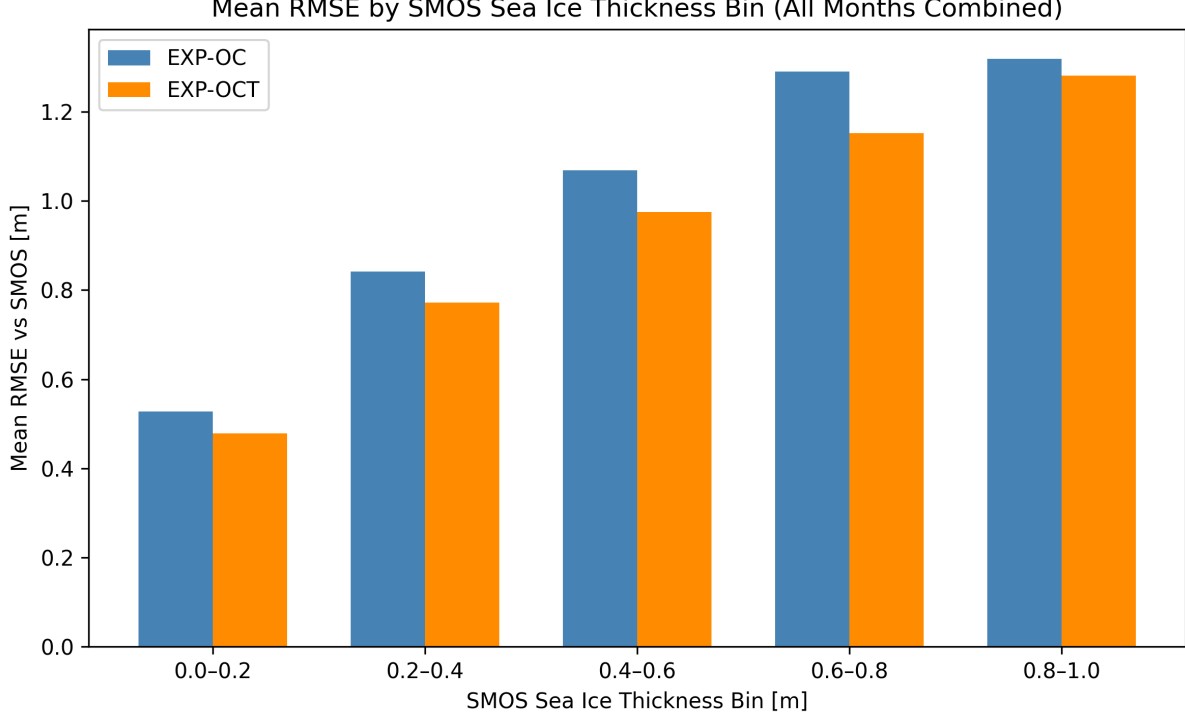

**Figure 5.** RMSE of modelled sea ice thickness from EXP-OC and EXP-OCT simulations compared to SMOS satellite observations, binned by SMOS sea ice thickness (0–1.0 m) taken across April–October for 2010–2022. Each bar represents the mean RMSE within the corresponding SMOS thickness interval.

positive trend, as well as in other regions where the trend is negative. EXP-OCT is a better match to the observed trends, increasing the pattern correlation from 0.37 to 0.5. In particular, EXP-OCT captures the negative trends observed in the Indian and West Pacific regions, while also exhibiting the strong positive trend seen in the Ross Sea.

The detrended interannual variability of pan-Antarctic SIE and SIV for EXP-OC and EXP-OCT is shown in Figure 9 at the
seasonal minimum and maximum in February and September. MSSS and ACC values for SIE over both the pan-Antarctic and individual regions are given in Table 1. Results vary substantially between regions and months, with a modest overall increase in pan-Antarctic MSSS and ACC for both February and September. The largest improvements occur in the Weddell, Ross, and Amundsen–Bellingshausen Seas, where sea ice is thickest and LEGOS data are most reliable. In contrast, performance decreases in the West Pacific and Indian sectors, where sea ice is thinnest and uncertainty in LEGOS SIT is greatest. EXP-OC
and EXP-OCT show some skill in reproducing the observed variability in the OSTIA SIC observations, though at a lower amplitude. EXP-OC and NorESM generally show minimal interannual SIV variability, whereas EXP-OCT exhibits larger variability, including two notable events: a pronounced increase in 2013–2014, followed by a sharp decrease in 2023. NorESM demonstrates limited skill in reproducing observed variability compared with EXP-OC and EXP-OCT for both SIE and SIV.





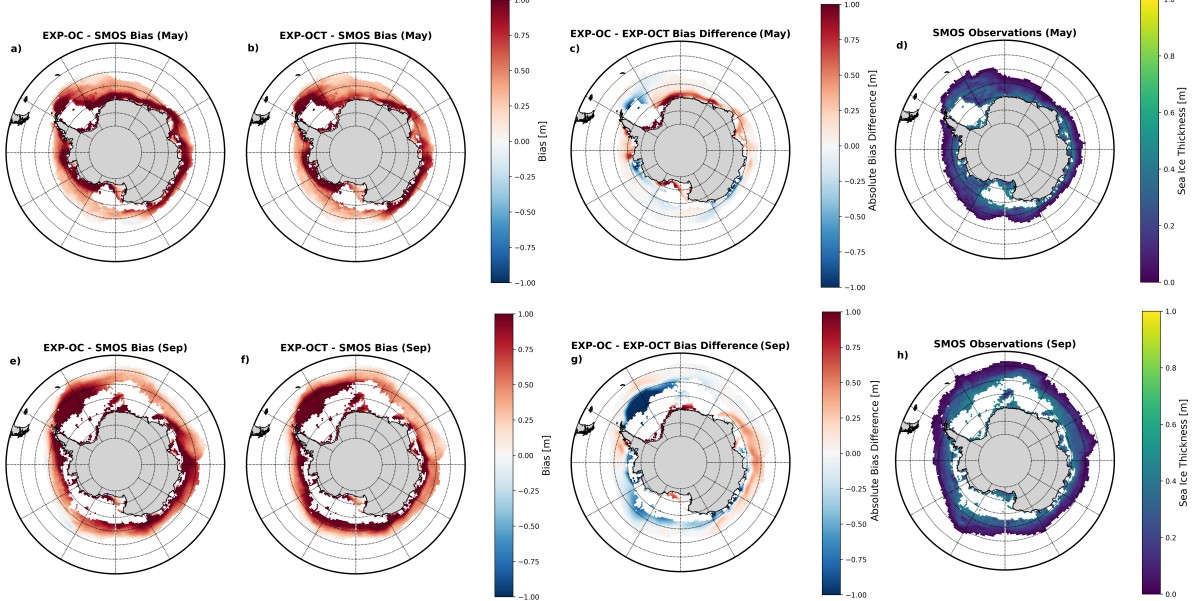

**Figure 6.** Spatial biases in climatological monthly SIT for May and September in EXP-OC, EXP-OCT and LEGOS compared to SMOS and monthly SIT climatology for SMOS for the 2010-2022 period. For the bias reduction plots, red (blue) indicates a reduction (increase) in bias in EXP-OCT. Grid cells where SIT is above 0.5 m in SMOS are masked out.

As expected, the spread in both SIE and SIV is reduced in EXP-OC and EXP-OCT relative to NorESM due to data assimilation
reducing uncertainty.

Errors in SIE may compensate for over- and underestimations of the true SIC. The IIEE (Goessling et al., 2016) provides a more robust measure by revealing the net displacement of the predicted ice edge relative to observations. The monthly climatology of pan-Antarctic IIEE (Figure 10) shows that SIT assimilation overall benefits ice edge skill from June to December, with notable improvement towards the end of the year, while from January-March IIEE is increased in EXP-OCT. Surprisingly,
the West Pacific where ice is thin and SIT assimilation has a weaker impact, contributes most to the reduced IIEE over the pan-Antarctic. The increased IIEE in EXP-OCT from January-March in the pan-Antarctic is caused by the Indian sector and the Weddell Sea. This increase in IIEE at this time is likely from thicker ice created by the SIT assimilation in the Weddell Sea, some of which advects into the Indian sector during the summer when sea ice velocity is at its highest. The reduction in IIEE during the last six months of the year is promising for the predictions, including in the West Pacific, where NorCPM has
previously had poor prediction skill (Xiu et al., 2025).

## 4.2 Prediction

We assess Antarctic SIE, SIT and SIC prediction skill using detrended anomaly correlation coefficients (ACC) in Figures 11, 12 and 13.




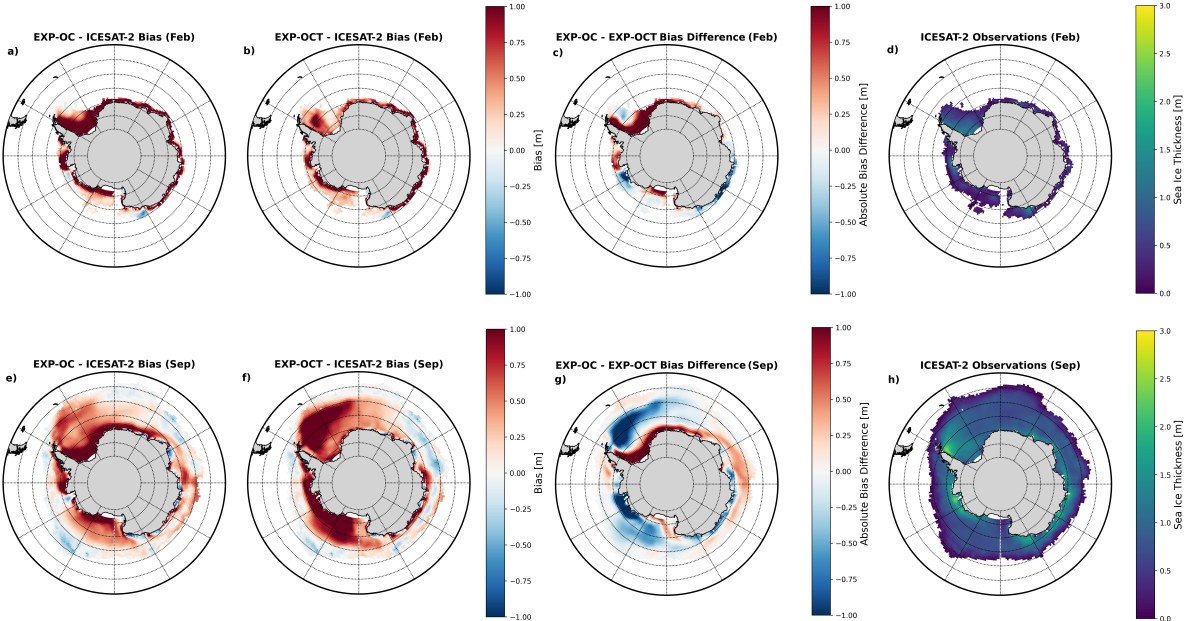

**Figure 7.** Spatial biases in climatological monthly SIT for February and September in EXP-OC and EXP-OCT, the bias difference between EXP-OC and EXP-OCT, and LEGOS compared to ICESat-2. Data are shown relative to the monthly SIT climatology from ICESat-2 for the 2019–2023 period. For the absolute bias reduction plots, red (blue) indicates a reduction (increase) in bias in EXP-OCT.

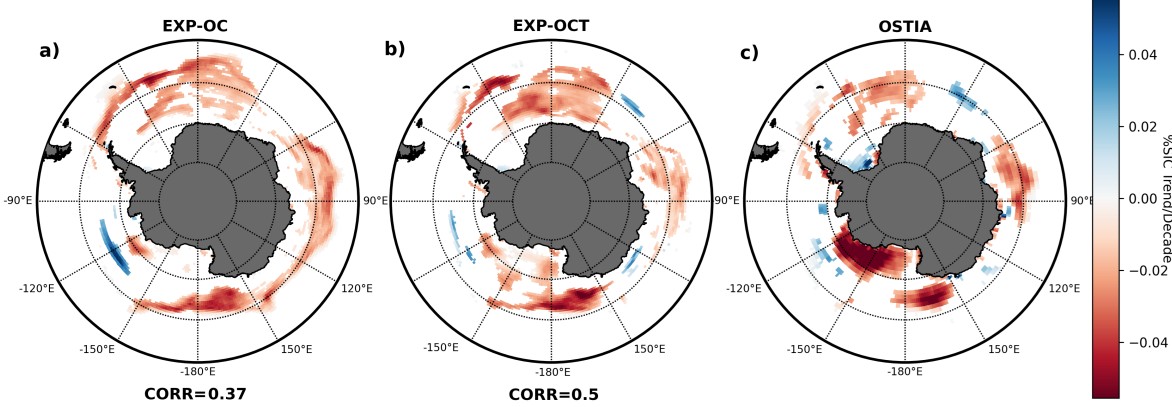

**Figure 8.** Linear trend of monthly SIC anomalies from 1995 to 2023 in EXP-OC, EXP-OCT and OSTIA in fractional sea ice concentration (normalised from 0 to 1). Only trends significant at a 95% confidence level are included in the plot. The spatial correlation pattern (referred to as corr) with trends from OSTIA trends is shown underneath the EXP-OC and EXP-OCT plots respectively.





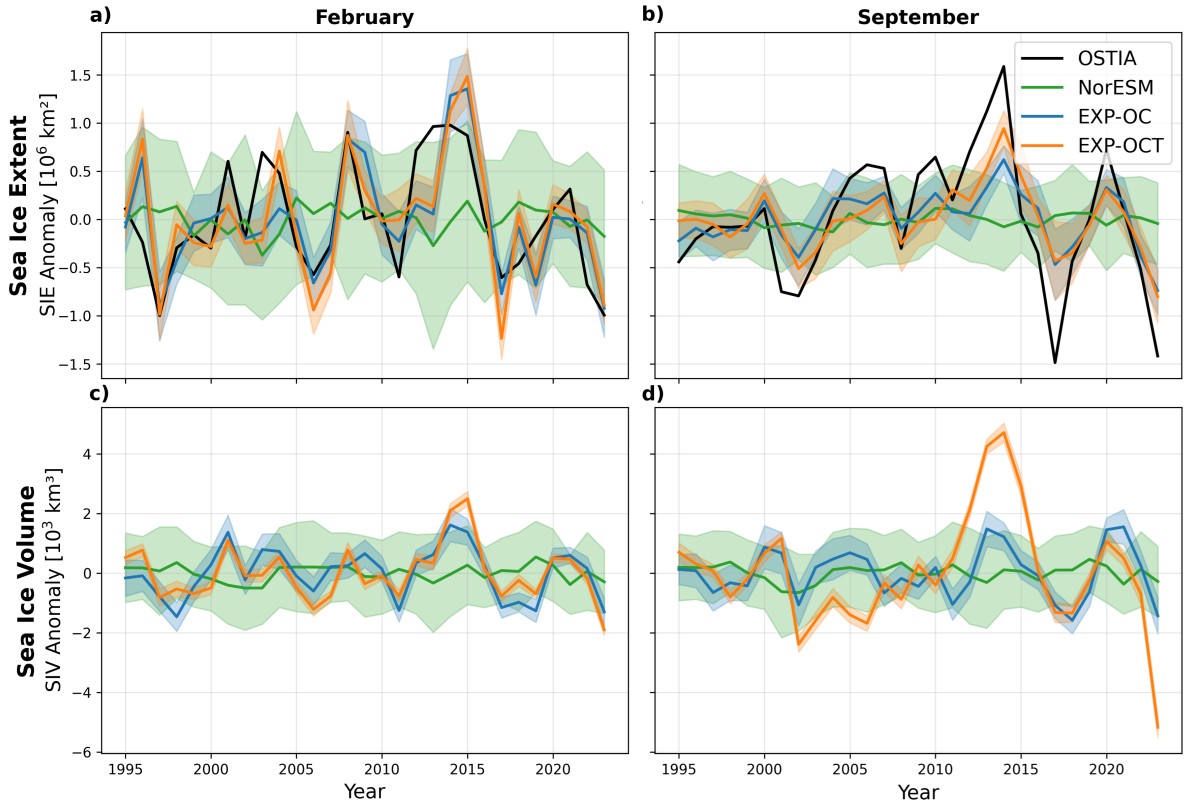

**Figure 9.** Time series of February and September detrended monthly anomalies in pan-Antarctic SIE (a, b) and SIV (c, d) between 1995 and 2023 for OSTIA, NorESM, EXP-OC and EXP-OCT. Shading shows ensemble spread for NorESM, EXP-OC and EXP-OCT defined as plus/minus one standard deviation.

In the West Pacific sector, SIE prediction skill in Exp-OC is generally low, while SIT skill remains skillful longer – i.e., SIE

skill is limited during the first half of the year, while SIT skill can extend until the end of the year from August to December. SIT assimilation greatly improves the prediction skill of SIT, which now extends to a 12-month lead time for all seasons except the austral winter (July-August). This leads to a notable improvement in SIE that now extends up to 9 lead months in the austral summer. A reduction in IIEE during the austral spring (Figure 10) likely contributes to enhanced skill during the subsequent growth season.

In the Ross Sea, the primary benefits of SIT initialisation are not as clear, and there seems to be a degradation for SIT prediction, particularly in austral winter, when a degradation was found with other SIT products (e.g., Figure 7). This region is also challenging, being highly dynamic and strongly seasonal (Raphael et al., 2019; Bushuk et al., 2021). In the austral summer and autumn (January-May), SIT was showing some improvement in the reanalysis, and the SIT prediction skill is more consistent up to five lead months, which also translates into improved SIE prediction skill during that time. SIE prediction skill

from the October initialisation is significantly enhanced, with skill extending up to 10 months.





**Table 1.** MSSS and ACC for EXP-OCT and EXP-OC with NorESM as reference for EXP-OC and EXP-OCT SIE, both against detrended anomaly OSTIA SIE over the reanalysis period for the pan-Antarctic and each of the five regions. For ACC, the system with the best performance in February and September is highlighted in bold. A-B Seas refers to the Amundsen and Bellingshausen Seas.

| MSSS | | | | |
|---|---|---|---|---|
| Region | Feb EXP-OC | Feb EXP-OCT | Sep EXP-OC | Sep EXP-OCT |
| West Pacific | **0.46** | 0.41 | **0.14** | 0.02 |
| Ross Sea | -0.19 | **0.20** | 0.22 | **0.25** |
| A-B Seas | -0.27 | **0.45** | 0.33 | **0.45** |
| Weddell Sea | 0.31 | **0.44** | 0.31 | **0.51** |
| Indian | **-0.19** | -0.28 | **0.43** | 0.42 |
| Pan-Antarctic | 0.06 | **0.14** | 0.25 | **0.30** |

| ACC | | | | |
|---|---|---|---|---|
| Region | Feb EXP-OC | Feb EXP-OCT | Sep EXP-OC | Sep EXP-OCT |
| West Pacific | **0.79** | 0.73 | **0.64** | 0.28 |
| Ross Sea | 0.15 | **0.45** | 0.68 | **0.72** |
| A-B Seas | 0.33 | **0.41** | 0.69 | **0.88** |
| Weddell Sea | 0.79 | **0.85** | 0.60 | **0.88** |
| Indian | **0.64** | 0.52 | 0.82 | **0.84** |
| Pan-Antarctic | 0.50 | **0.65** | 0.71 | **0.73** |

EXP-OCT shows good improvement over EXP-OC for both SIE and SIT skill in the Amundsen and Bellingshausen Seas, with higher ACC values sustained over longer lead times, particularly for winter and spring targets. SIT initialisation enhances the representation of initial ice volume and melt timing, leading to improved prediction performance. EXP-OCT exhibits stronger SIC skill improvement in January from October initialisation (Figure 13). However, the increase in Ross Sea SIE skill
does not correspond with higher SIT skill, which may suggest that improved sea ice conditions in the adjacent Amundsen Sea then later advect into the Ross Sea (Falco et al., 2024), which enhances SIC predictions without improving local SIT skill. Transported or newly grown ice may melt, deform, or interact with existing ice in ways not accurately represented by the model.

The Weddell Sea (Figure 12) exhibits the most pronounced increase in SIT prediction skill thanks to SIT assimilation.
SIE improvements coincide well with the seasonal timing of the skill but are slightly lower than for SIT. EXP-OCT extends significant SIE predictability by approximately three months from the January start time. Figure 13 shows ACC increases in both SIC and SIT in the Weddell Sea for up to nine lead months, though SIC ACC weakens near the ice edge, consistent with smaller basin-wide SIE skill gains. Xiu et al. (2025) identified atmospheric initialisation as an important mechanism during austral spring in the Weddell Sea, whereas we demonstrate that SIT initialisation is the key mechanism governing SIT
predictability during austral summer and autumn.





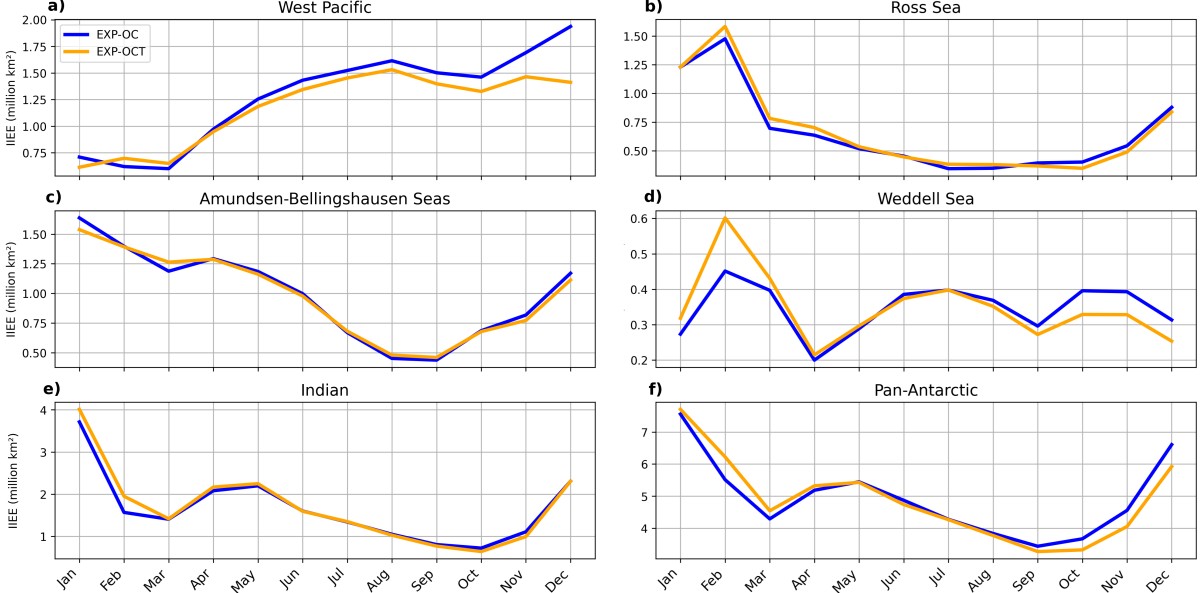

**Figure 10.** Monthly climatologies of the Antarctic IIEE in each of our defined regions, and for the pan-Antarctic, for EXP-OC and EXP-OCT compared with OSTIA observations between 1995 and 2023. Note that the y-axis scale differs in each subplot in order to show differences in each region clearly.

In the Indian Ocean sector, the impact from SIT is marginal. SIT is only slightly improved on the October forecast, and the prediction skill is modest. SIE skill is degraded from the January start time in EXP-OCT. The thinner, more variable sea ice cover and stronger oceanic influence in this region likely limit the impact of SIT initialisation, especially in summer when there is very little ice cover in this region.

Basin-wide, EXP-OCT produces higher and more persistent ACC values than EXP-OC for SIT, although many differences are not statistically significant. Predictive skill increases markedly from the October initialisation, with both SIT and SIE skill remaining significant at lead times approaching one year. In January, there is an additional SIE skill for up to three months, though SIT improvement is less clear.

Regions showing enhanced SIT skill generally exhibit corresponding SIE improvements, indicating that assimilating SIT 375  strengthens the model's initialisation of ice volume. This enhances the persistence of sea ice anomalies and improves both SIE and SIT forecasts. At short lead times, both experiments perform similarly; however, at longer leads, predictive skill in EXP-OC decays more rapidly, while EXP-OCT retains skill in both SIC and SIT. As shown in Figure 13, EXP-OCT exhibits strong initial SIT skill across the Antarctic, with SIC skill increases that persist at longer lead times from October. SIT and SIC prediction skills in EXP-OCT decline slightly toward the end of the melt season, but typically strengthen again in autumn and 380  persist through winter.





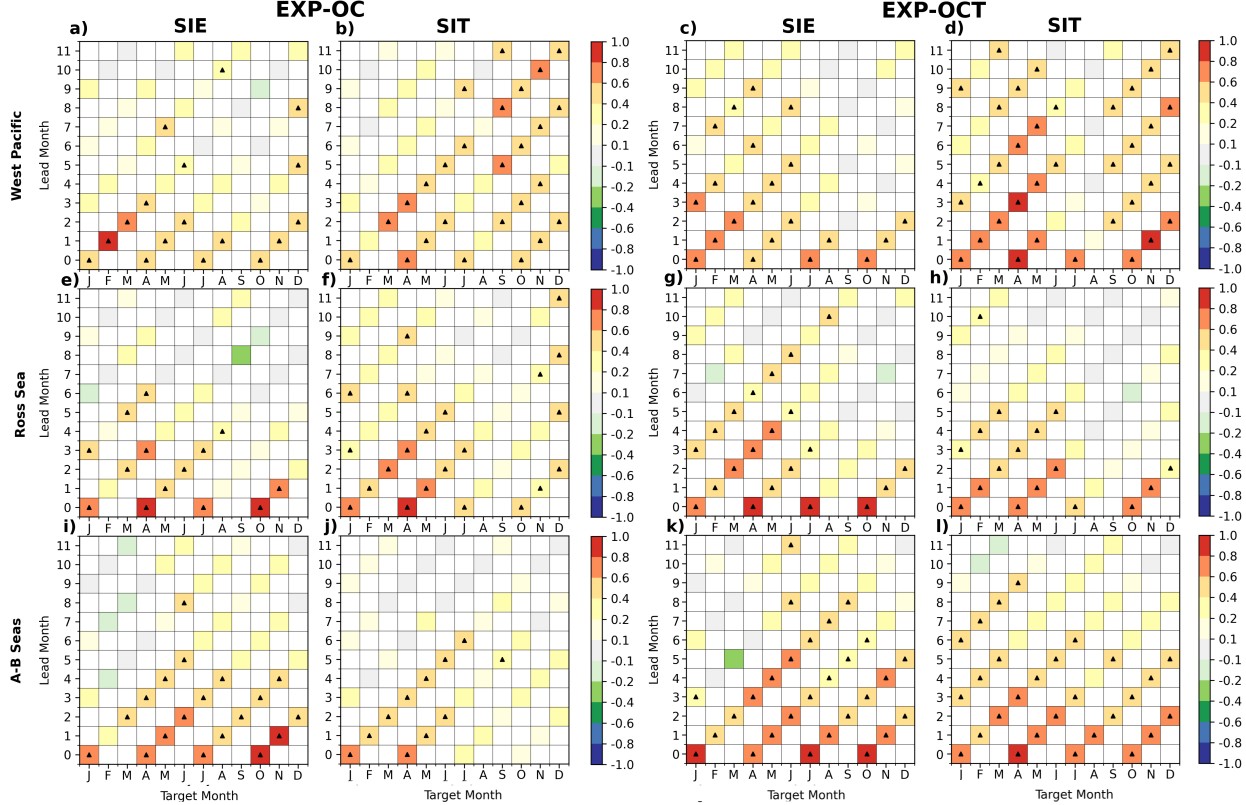

**Figure 11.** ACC for the West Pacific, Ross Sea and Amundsen-Bellingshausen Seas SIE and SIT for our seasonal hindcast datasets for EXP-OC and EXP-OCT in comparison with OSTIA SIC and LEGOS SIT. Target months are shown on the x-axis, with lead months shown on the y-axis. Triangle markers represent statistically significant (95%) ACC values.

Overall, the results demonstrate a robust link between SIE and SIT predictive skill. Parallel improvements are evident—for example, during austral spring in the Amundsen–Bellingshausen Seas, austral autumn in the West Pacific, and in pan-Antarctic forecasts from the October initialisation. Possible evidence for a prediction barrier for SIT emerges in August–September, preceding a corresponding barrier for SIE in the austral spring. This suggests that inadequate SIT estimates at the end of 385 winter are associated with poor SIC and SIE estimates at the end of summer. However validation of the reanalysis for SIT with independent SIT products indicates that LEGOS SIT may have increased SIT too much during the peak sea ice volume period which may cause this barrier in our model. Figure 13 further shows great improvements in SIT ACC at initialisation that persist through the forecast period, though they weaken slightly at three months but re-emerge at six lead months. SIC improvements are weaker but remain evident, particularly in the Ross and Weddell Seas.

The West Pacific sector is the only region that departs from this pattern, possibly because LEGOS observations reveal persistent pockets of thick coastal ice that are better represented in EXP-OCT, as indicated by the reduced bias in Figure 3. Nevertheless, the model underestimates SIT growth relative to observations, which limits SIT predictive skill at the end




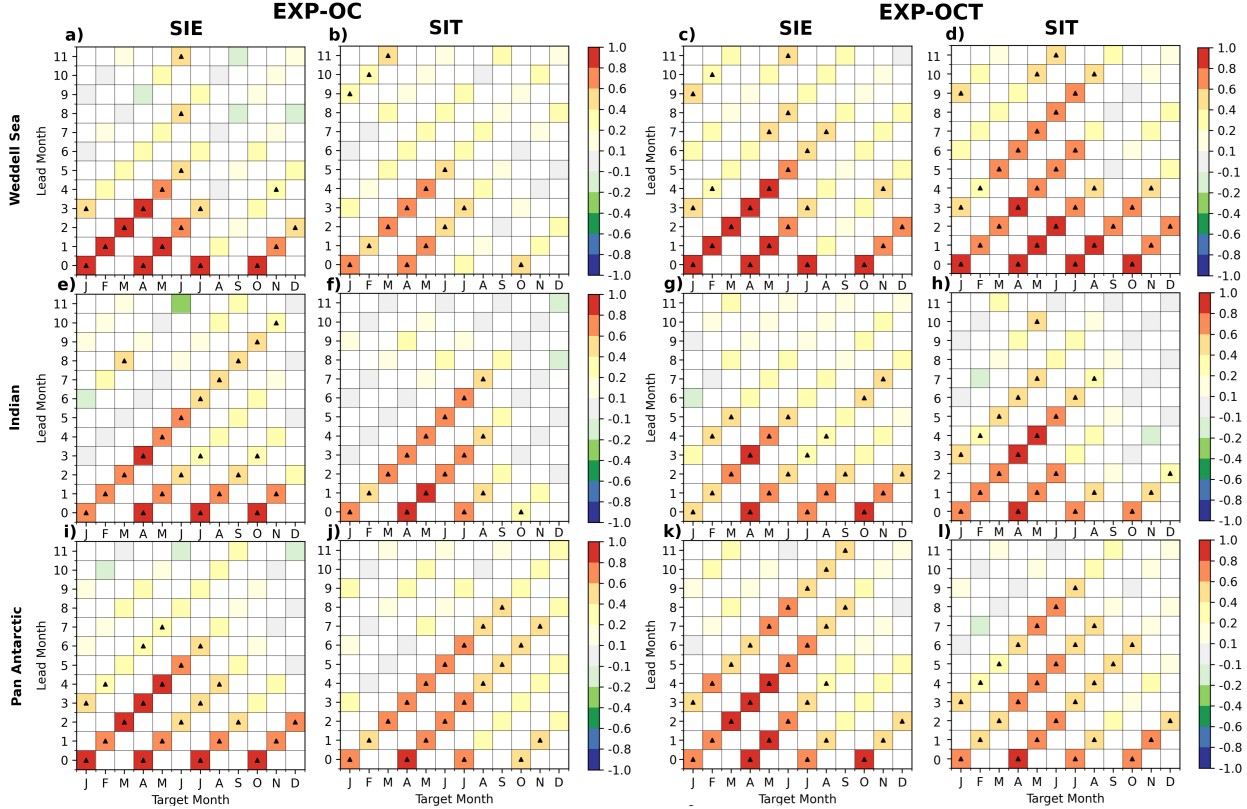

**Figure 12.** ACC for the Weddell Sea, Indian Sector and pan-Antarctic SIE and SIT in EXP-OC and EXP-OCT in our seasonal hindcast datasets for EXP-OC and EXP-OCT in comparison with OSTIA SIC and LEGOS SIT. Target months are shown on the x-axis, with lead months shown on the y-axis. Triangle markers represent statistically significant (95%) ACC values.

of the growth season and subsequently degrades SIE predictability through spring and summer in experiments without SIT initialisation.

## 5  Discussion and conclusions

This study demonstrates that assimilating sea ice thickness (SIT) observations substantially improves the reanalysis and prediction of Antarctic sea ice in a global coupled climate model. The inclusion of SIT data enhances the seasonal cycle and variability of sea ice, particularly in the West Antarctic. Improvements are most pronounced in the Ross, Weddell and Amundsen–Bellingshausen Seas, where SIT assimilation leads to better estimates of summer sea ice minima and extended lead time skill. In the West Pacific, a notable increase in January–June prediction skill in EXP-OCT corresponds to reduced IIEE between September and December, where NorCPM previously showed little predictive capability (Xiu et al., 2025). The





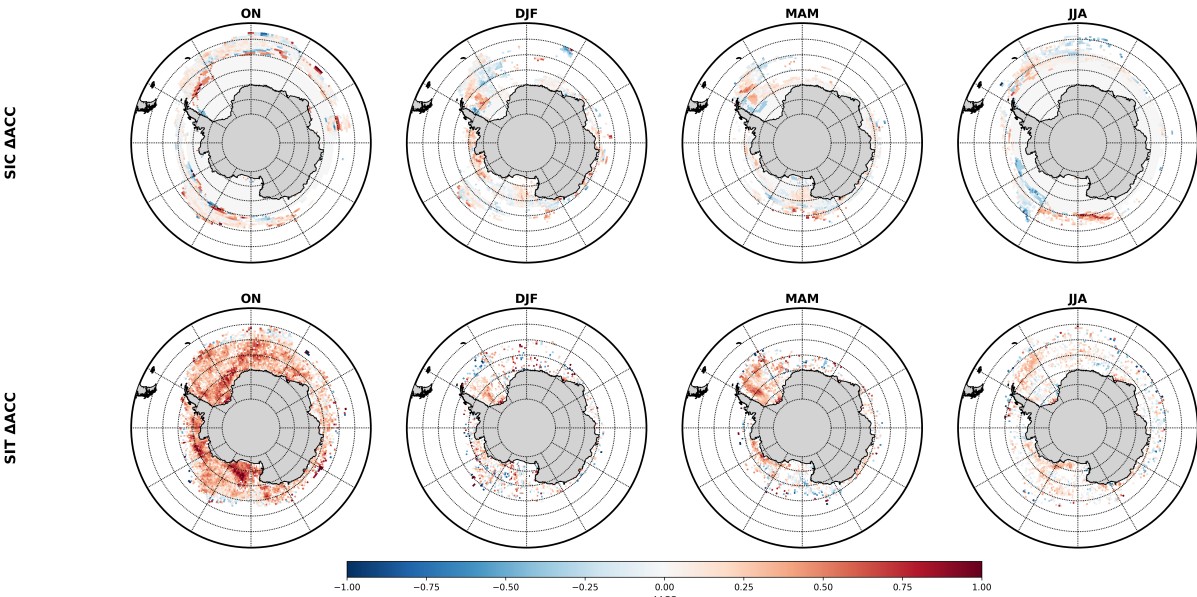

**Figure 13.** ACC difference between EXP-OCT and EXP-OC for Antarctic SIC (top) and SIT (bottom). Results are shown for forecasts initialised in October for seasonal averages in ON, DJF, MAM and JJA. Positive values (red) indicate higher skill in EXP-OCT relative to EXP-OC, and negative values (blue) indicate lower skill. Only ACC values which are statistically significant at the 95% confidence level according to a two-tailed t-test are plotted.

enhanced predictability stems from the increased persistence of SIT anomalies, resulting in better estimates of ice melt timing and ice growth.

SIT assimilation generally improves predictive skill across most Antarctic regions, although limited or negative effects are
found in parts of East Antarctica, particularly in the Indian Ocean sector, suggesting regional dependencies in the response to SIT initialisation. An end-of-winter barrier in SIT predictability, possibly linked to the austral spring barrier in SIE prediction, highlights ongoing challenges in representing seasonal transitions. However this barrier may also reflect limitations in the LEGOS SIT dataset. Nevertheless, the overall improvement in both SIT and SIE prediction skill demonstrates the value of incorporating newly developed SIT observations into Antarctic forecasting systems.

This study demonstrates that the long-term LEGOS SIT dataset provides valuable information for sea ice reanalysis through data assimilation. However, its uncertainties require careful characterisation, particularly in regions close to the ice edge. Compared to SMOS and ICESat-2, the LEGOS SIT dataset generally indicates a thicker Antarctic sea ice cover. SMOS SIT is known to exhibit a thin-ice bias, while ICESat-2 offers substantially more limited temporal coverage. Despite these differences, the three datasets show strong complementarity and exhibit consistent spatial patterns with the EXP-OCT reanalysis, particularly
in the spatial distribution pattern of the sea ice thickness.



This work represents the first long-term assessment of Antarctic SIT assimilation over a multi-decadal hindcast period, establishing SIT as a key mechanism for improving Antarctic sea ice prediction. Future developments should focus on combining SIT assimilation with atmospheric nudging, which can enhance prediction skills in the Indian sector and Weddell Sea (Xiu et al., 2025). Additionally, incorporating other SIT products such as SMOS and ICESat-2 can further enhance the SIT initialisation,

benefiting from the strengths of each product. These advances will not only enhance the skill of Antarctic sea ice forecasts but also provide critical support for navigation, operations, and climate monitoring in a region experiencing rapid environmental change and increased human activity (Znój et al., 2017; Hou et al., 2025).

*Code and data availability.* During the revision phase, NorCPM data and scripts used in this study will be uploaded to the NIRD

Research Data Archive which is the Norwegian long-term storage service for research data under the Open-Access license. EN4.2.1 and EN4.2.2 salinity and temperature profile data can be downloaded from https://www.metoffice.gov.uk/hadobs/en4/download-en4-2-1.html, OSTIA SST and SIC data is available from https://data.marine.copernicus.eu/product/SST_GLO_SST_L4_REP_OBSERVATIONS_010_011/description and https://data.marine.copernicus.eu/product/SST_GLO_SST_L4_NRT_OBSERVATIONS_010_011/description, LEGOS SIT data for the Arctic and Antarctic is available from https://www.legos.

omp.eu/ctoh/catalogue/?uuid=c52f78d1-a658-4c56-a66c-459de0d88fb6. SMOS Antarctic SIT data is available from ftp://ftp.awi.de/sea_ice/product/smos/v3.3/sh/. ICESat SIT data is available from https://doi.org/10.6084/m9.figshare.28899965.v1.

*Author contributions.* FC, YW and NW all wrote code pertaining to sea ice thickness assimilation in NorCPM. YW and FC wrote code pertaining to assimilation of sea ice concentration and ocean observation assimilation in NorCPM. NW designed and conducted the experiments. NW produced the figures and the paper with assistance, feedback and edits from YW and FC.

*Competing interests.* The authors declare that they have no competing interests.

*Acknowledgements.* The authors thank Yongwu Xiu from the China Meteorological Administration for helpful discussion and feedback. This study was supported by the Norges Forskningsråd (Grant Nos. 328886, 350390, 309562, 352204). It was also partly funded by ObsSea4Clim "Ocean observations and indicators for climate and assessments" funded by the European Union. Grant Agreement number: 101136548. DOI: 10.3030/101136548. This work also received grants for computer time

from the Norwegian Program for supercomputer (NN9039K) and storage grants (NS9039K).



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
