# Peer review of "Enhanced Predictability of Antarctic Sea Ice through Sea Ice Thickness Assimilation"

_EGUsphere, 2025_

## Referee Comment (RC1)

**Review of "Enhanced Predictability of Antarctic Sea Ice through Sea Ice Thickness Assimilation" by Nicholas Williams et al.**

**Summary**

Estimating Antarctic sea ice thickness (SIT) is essential for understanding sea ice mass balance and improving prediction skill, yet it remains challenging in Earth system models. In this study, the authors use NorCPM to produce two reanalysis datasets, one of which assimilates a 30-year LEGOS SIT product. These reanalyses are then used to initialize seasonal hindcasts from 1995 to 2022, with start dates in January, April, July, and October. The resulting sea ice estimates and predictions are evaluated against available observation-based datasets.

Overall, this manuscript makes a timely and valuable contribution to seasonal-to-interannual sea ice prediction and will be of interest to the cryosphere communities. While the study presents promising results, several aspects require further clarification and improvement. I therefore recommend minor revisions and believe that addressing these points would enhance its suitability for publication.

**General Comments**

1. The manuscript frequently uses the term "enhanced predictability", while the analyses mainly demonstrate improved prediction skill resulting from better initialization through SIT assimilation. Since predictability refers to the intrinsic limits of the climate system, whereas prediction skill reflects model performance, the authors should clarify which aspect is improved. If the results primarily indicate enhanced prediction skill in NorCPM, the terminology should be revised accordingly throughout the manuscript, including the title.

2. Line 111–112: NorESM is run with CMIP5 historical forcings and RCP8.5 beyond 2005. Given the citation of Xiu et al. (2025), which shows that improved atmospheric state representation can enhance sea ice prediction skill, the authors are encouraged to discuss whether the use of RCP8.5 forcing may introduce systematic atmospheric biases, and whether improvements in atmospheric forcing (rather than nudging) could lead to more realistic atmospheric states and improved sea ice prediction skill.

3. The assimilation implementation requires clarification. In Lines 128–131, it is unclear whether SIC is updated twice within the same assimilation cycle, and if so, how these updates are implemented and treated differently. In addition, both the R-factor and K-factor are used to inflate the observation error; please clarify their respective roles and, if possible, quantify their effects, for example by indicating how much the ensemble spread or observation error is inflated.

4. Figure 4: Compared with SMOS and ICESat-2, the assimilated LEGOS SIT appears

to have a larger RMSE than EXP-OC. If so, this may indicate better agreement of EXP-OC with these independent datasets. The authors are encouraged to clarify this by directly comparing both EXP-OC and LEGOS SIT against SMOS and ICESat-2, and to explain why further assimilating LEGOS SIT nonetheless leads to improved sea ice estimates and prediction skill.

**Minor Comments**
1. Line 47: Please clarify what "OISST" refers to at first mention?

2. Line 80: "ICESAT-2" should be corrected to "ICESat-2."

3. Only one third of the available ensemble members is used for prediction. Please clarify the rationale for this choice and discuss whether the reduced ensemble size could affect prediction skill.

4. Section 3.2: This section requires clarification. *bfRMSE* is mentioned but not used elsewhere; please clarify or remove it. In addition, define *MSE* and explain how *MSE$_{forecast}$* and *MSE$_{reference}$* are computed. In Equation (4), the variables appear to represent anomalies rather than the raw model and observation values described in Line 241.

5. Line 275 and Figure 4: EXP-OCT does not consistently exhibit lower RMSE than EXP-OC in July–September; please revise the corresponding text. In addition, the comparison and related discussion should refer to ICESat-2 rather than ICESat.

6. Figure 3: Compared with LEGOS SIT, EXP-OC overestimates sea ice prior to SIT assimilation, while EXP-OCT shows an underestimation afterward, particularly in the Weddell Sea. Please discuss the potential mechanisms underlying this shift.